# Decoding Semantics: A Multi-Modal CNN as a Model for Human Literacy Acquisition

**Tommy Clausner**[1,2*]**, Ole Jensen**[1,2]

[1]Oxford Centre for Human Brain Activity, Oxford Centre for Integrative Neuroimaging,
Department of Psychiatry, University of Oxford, Oxford, OX3 7JX, UK

[2]Department of Experimental Psychology, University of Oxford, Oxford, OX2 6GG, UK

## Abstract

**While visually presented objects (e.g. a picture of a rat) and words (e.g. the word *rat*) appear perceptually different, they evoke a similar semantic activation in the human brain. A key question in understanding human reading acquisition is how semantic representations emerge such that visual object representations and written words are meaningfully linked. We used a convolutional neural network (CNN), trained such that both object images and written word stimuli activate the same output unit. Our findings indicate that cross-modal semantic representations emerge gradually across layers. Using representational similarity analysis of the layer activations, we further were able to show that incongruent information affects the network's performance via interfering projections to a high-dimensional space. This suggests that the acquisition of literacy can be modelled as the projection of object and word features, processed via the same neuronal substrate - the visual cortex - into a shared semantic space. Our approach offers a new avenue to uncover the neuronal substrate of human literacy acquisition by using representational similarity analysis to link representations in the CNN to brain imaging data.**

**Keywords:** Semantics; Multimodal; CNN; Reading

## Introduction

Understanding how humans acquire and process written language is a fundamental question in cognitive neuroscience. A key element of this process is the ability to process both objects and written or printed words, using shared neural structures in the visual cortex (Dehaene et al., 2005), and assign a joint meaning to both representations. The "visual word form area" (VWFA), located in the left occipitotemporal lobe, has been proposed as a specialised region involved in recognising written words (Cohen et al., 2000; Dehaene & Cohen, 2011). Other specialised "processor"-areas like the fusiform face area (FFA) for human face processing (Kanwisher & Yovel, 2006) have been suggested to reflect a similar purpose on a different visual modality. Those findings promote a functionally and anatomically modular view of the human brain, where semantic representations are formed via a coordinating hub region (Patterson et al., 2007). However,

more recent theories argue that information is rather represented in a high-dimensional shared representation space, into which patterns of information - or features - are projected, rejecting the idea of canonical, anatomically defined subsystems (Haxby et al., 2020). We hypothesise that if this is the case, a single convolutional neural network (CNN) should be able to represent multiple visual categories - objects and written words - simultaneously. Those representations are hypothesised to manifest as separate feature projections from the convolutional layers to shared representations in the fully connected layers.

A more recent model that explicitly links words and images on the basis of *meaning* is the CLIP model. It consists of a vision and a language encoder that are jointly trained via a shared representation matrix into which both encoders project (Radford et al., 2021). This model was utilised to explain first word acquisition in a single infant to which a camera was attached and in which the parent's interactions (i.e. naming objects) and the visual scene were related (Vong et al., 2024). However, the joint training was performed by abstracting the auditory information into "digital" words that are biologically implausible. In addition, since vision and word encoders were entirely separate, one could argue that this procedure merely emulates self-supervised labelling, which implies that the words themselves are not linked to the visual representation of the named objects. In the domain of reading, the human brain utilises only one encoder (the visual cortex) for object and written word recognition. Recent studies, employing CNNs, demonstrated that a structure akin to the VWFA can emerge within a CNN's structure, simply by joint multimodal training (Hannagan et al., 2021; Agrawal & Dehaene, 2024). However, again, investigating the acquisition of *semantic* knowledge (i.e. the link between object and written word features) remains to be investigated. Since the acquisition of reading cannot be explained on the basis of evolution but rather relies on utilising evolutionary older structures (Dehaene & Cohen, 2007), semantics as "the literal meaning of words" (Kearns, 2017), are likely to be found embedded within other neuronal subsystems. Providing a compelling explanation on how "neuronal recycling" (Dehaene, 2005) can lead to the formation of semantic representations in a high-dimensional projection space could help to form a deeper understanding of human reading acquisition.

---

*Corresponding author: tommy.clausner@psych.ox.ac.uk

In the present paper, we demonstrate that semantic representations emerge even in very simple CNNs by the joint learning of visual representations of objects and their respective written word forms. We trace their respective relationship throughout the model and demonstrate how the abstractions of visual features, processed using a single encoder, are projected into a shared semantic representation space.

## Methods

### Network Architecture

Previously, it has been argued that CNNs can be used as models for the human visual cortex (Kriegeskorte, 2015). Here, we used VGG 11 (Simonyan, 2014), which was motivated threefold: Firstly, compared to widely considered more biologically plausible CORnet-Z (Kubilius et al., 2018), VGG 11 performs better on image recognition tasks when simultaneously trained on image and word data. Agrawal & Dehaene (2024) reported 36.8% accuracy using CORnet-Z for ImageNet data in an otherwise highly similar experimental setup. Second, VGG style networks still remain comparable to the hierarchical processing of the human visual system, with brain likeliness scores similar to, or exceeding the CORnet family (Nonaka et al., 2021). Although the most recent version, CORnet-S (Kubilius, Schrimpf, Kar, et al., 2019), outperforms VGG style networks on the "Brain-Score" metric with comparable ImageNet classification results (Schrimpf et al., 2018), it performs less well on the "brain hierarchy score" (Nonaka et al., 2021). Third, the recurrent connections in e.g. CORnet-S require time-resolved training and analysis of layer activations, which adds computational cost and complexity. VGG 11 in turn is still simple enough to gain meaningful insight from a straight forward analysis of individual layer representations. The network was implemented as provided by PyTorch (Paszke et al., 2019) with one key modification: the number of units in the output layer was reduced from 1000 to 533 to match the number of classes in our dataset (see Figure 1 b).

### Model Training

Our primary objective was to approximate infant reading acquisition at the stage of first word learning. Before learning to read, infants are already capable of naming objects, which we approximated by a model pre-trained on ImageNet 1k (Deng et al., 2009). Afterwards, the model was further trained on the same dataset alongside a set of newly created word stimuli. We opted for supervised learning (as opposed to e.g. contrastive learning), again to approximate human literacy acquisition at an infant stage. Word stimuli were created from the human-readable ImageNet labels. Each word image was created by randomly choosing one of 143 different fonts, with a font size between $12$ and $48$ $pt$. Afterwards, each word was rotated by up to $\pm 30°$ and sheared by an up to $\pm 45°$ offset. Finally, random noise with an intensity between 0 and 50% was added. This procedure is similar to what was done previously (van Vliet et al., 2022). The noise background reflects noisy conditions (e.g. changes in lighting) mirroring the robustness of human word recognition. In addition, noise increases background complexity to better match the mixed modality testing stimuli (see below). See Figure 1 for an example. The data were split into 80% training data, 10% validation data and 10% test data. Since the word stimuli provide a much smaller set of features, all labels that contained white spaces, dashes or other non-letter characters were excluded to standardise the word images as much as possible. Furthermore, labels with more than ten characters were excluded to ensure that the respective word stimuli do not exceed the image dimensions too much, while still ensuring high variability in font style and size. This resulted in a final set of 533 classes, which we considered sufficiently many to not trivialise the problem. Each training batch consisted of a mix of object and word images. Thereby, the cross-entropy loss was computed identically, irrespective of whether the example stemmed from the ImageNet dataset or the word images.

The model's parameters were optimised using stochastic gradient descent (SGD) with a learning rate of $lr = 0.001$, a weight decay of $wd = 0.0005$ and a momentum of $m = 0.9$. A training scheduler was used to reduce the learning rate by a factor of $0.1$ on two consecutive plateauing epochs of validation loss. Training was stopped after nine epochs, reaching a final validation accuracy of 88.3%.

### Model Testing

We investigated semantic representations at the behavioural level, by exposing the network to conflicting stimuli it was not trained on. Furthermore, we tracked the formation of semantic information across different layers using representational similarity analyses (Kriegeskorte et al., 2008). To probe the model's semantic representations, two additional datasets were created: congruent and incongruent images. Congruent stimuli were created by overlaying the object images with the corresponding words, whereas incongruent stimuli were created by overlaying the object images with non-corresponding words. This is similar to the picture-word interference task (Rosinski et al., 1975). Each word was created similarly as the pure word stimuli, but without additional rotation or shearing to reduce the amount of variation in this test-only data. An on average $12$ $pt$ larger font size was chosen to boost the word size over the training set, to avoid that only tiny fractions of the combined stimuli were covered due to small font sizes in combination with small words.

After applying a softmax operation to the output layer, an example was classified as "correct" if the target unit had the highest probability (top 1), was among the five highest probability values (top 5), the 27 highest probability values (top 5%) or the highest 53 probability values (top 10%). In the case of incongruent stimuli, accuracy values were computed separately for the class corresponding to the object part or the word part.

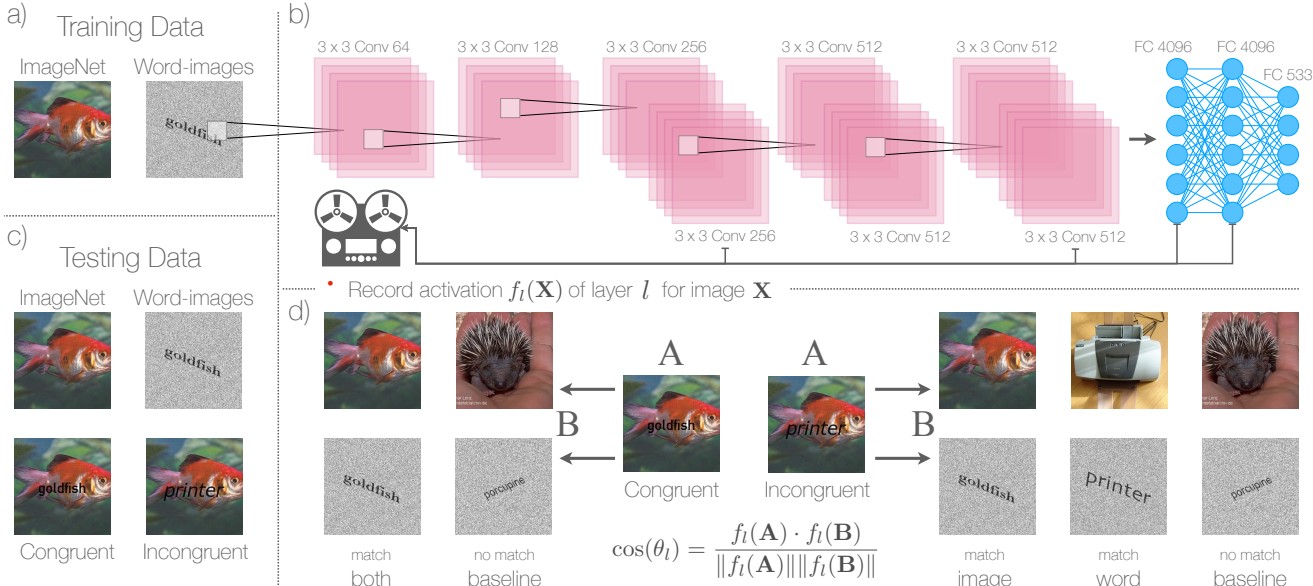

Figure 1: Overview of the methodology. **(a)** Training Data: The model was trained on 533 classes of ImageNet and corresponding written word images. **(b)** The VGG 11 architecture, consisting of eight convolutional layers (Conv) followed by two fully connected layers (FC). The final (third) FC layer was modified to have 533 output units to match the number of classes. **(c)** Testing Data: The testing phase involved ImageNet and written words, as well as congruent and incongruent mixed modality stimuli. **(d)** Experimental Procedure: During testing, activations $f_l(\mathbf{X})$ for each layer $l$ for stimulus $\mathbf{X}$ were recorded. The cosine similarity, $\cos(\theta_l) = \frac{f_l(\mathbf{A}) \cdot f_l(\mathbf{B})}{\|f_l(\mathbf{A})\|\|f_l(\mathbf{B})\|}$ was computed, between the activations of congruent or incongruent images (**A**) and the activation of object-only or word-only testing stimuli (**B**). Object or word parts of congruent or incongruent stimuli could either match the respective object-only or word-only image categories or not (baseline). This means that for congruent stimuli, four separate cosine similarity values are computed, whereas for incongruent stimuli, six separate cosine similarity values are computed.

## Representational Similarity Analysis

Representational similarity analysis (RSA) is a widely used method in neuroimaging, to assess the similarity of brain response patterns across different conditions (Kriegeskorte et al., 2008). Here, we applied RSA to investigate the similarity of layer activations of the network with respect to different input stimuli. During testing, the activation of the fourth convolutional layer (mid conv), the last convolutional layer (last conv) and both fully connected hidden layers were recorded for each individual testing image. We focused our analysis on middle and deeper layers, since early convolutional layers are known to primarily encode simple features like edges and textures, while deeper layers capture more complex structures or concepts (Zeiler & Fergus, 2014; Yosinski et al., 2015). Semantic representations were expected to be found in the fully connected layers (Qin et al., 2018).

The representational similarity (RS) was computed as the cosine similarity between the activation $f$ of two stimuli **A** and **B** in a given layer $l$: $\cos(f_l(\mathbf{A}), f_l(\mathbf{B}))$. Since it is highly unlikely that each single image is represented exactly orthogonal throughout the network, we also expected some RS with any image (e.g. because of shared features across classes). For this reason, the RS was computed relative to a baseline.

The baseline similarity applied to the RS between **A** and **B** $RS_l^{BL(\mathbf{A},\mathbf{B})}$ was computed as the average cosine similarity between the activations for **A** and $N = 1000$ randomly selected examples **C**:

$$RS_l^{BL(\mathbf{A},\mathbf{B})} = \frac{\sum\limits_{\mathbf{C} \in \Gamma}^{N} \frac{f(\mathbf{A}) \cdot f(\mathbf{C})}{\|f(\mathbf{A})\|\|f(\mathbf{C})\|}}{N} \quad (1)$$

, where $\Gamma$ is the set of all possible classes and **C** a randomly drawn example. The final RS between the activations of **A** and **B** was then computed as the difference to the respective baseline:

$$RS_l^{(\mathbf{A},\mathbf{B})} = \frac{f_l(\mathbf{A}) \cdot f_l(\mathbf{B})}{\|f_l(\mathbf{A})\|\|f_l(\mathbf{B})\|} - RS_l^{BL(A,B)} \quad (2)$$

This measure reflects the RS in a given layer that can solely be attributed to activation similarity between two classes over all other classes.

We conducted an RSA for the following combinations: object-only and word-only stimuli (and vice versa), congruent and object-only or word-only stimuli, as well as incongruent and object-only or word-only stimuli. In case of an incongru-

ent stimulus, RS analyses have been performed for the object and word part separately. See Figure 1 d) for example comparisons for congruent and incongruent stimuli, and the illustrations in Figure 3 for visual representation of the comparisons performed.

## Activation Similarity Maximisation

While RSA is a suitable method to estimate *where* and to which degree semantic representations are formed, it does not reveal *how* semantic representations are formed. A common approach to reveal how CNNs represent visual features is to apply activation maximisation (Erhan et al., 2009; Qin et al., 2018). In its simplest form, the pixel of an initially random stimulus are iteratively changed via gradient ascend, such that the output of a certain unit is maximised. This generates an image that leads to a very high activation at the desired output unit. Thereby, it can be revealed what the network considers a very prototypical representation of the respective target class.

Here, we would like to gain insight into how semantic representations are formed and thus the link between the visual representation of a word and the respective target class. For this reason, we introduced a multiplicand to regularise the growth of the activation $a$ at the output unit $i$. The loss function to generate this new stimulus has been reformulated as a minimisation problem becomes:

$$loss = -\frac{f_l(\mathbf{Y}) \cdot f_l(\mathbf{B})}{\|f_l(\mathbf{Y})\|\|f_l(\mathbf{B})\|} * a^{(i)} \qquad (3)$$

, with $f_l(\mathbf{Y})$ being the activation of the optimised stimulus $\mathbf{Y}$ in layer $l$, $f_l(\mathbf{B})$ the activation of the reference word stimulus in layer $l$ and $a^{(i)}$ the activation $a$ of unit $i$ in the output layer. Using this procedure, a stimulus was created using 500 optimisation steps that produced a high activation in the target unit, where activation patterns in layer $l$ similar to the activation of the reference stimulus are rewarded, and dissimilar representations are penalised. By using a word stimulus as the reference stimulus we obtain a new stimulus $\mathbf{Y}$ that is represented word-like within a given layer, but at the same time maximises the output for a certain class. Afterwards, we computed the RS between $f_l(\mathbf{Y})$ and $f_l(\mathbf{B})$ as a proxy for how well the word stimulus represents the target class in each layer.

To probe semantic processing, the class of the reference stimulus was either chosen corresponding to the output unit $i$ (within-class) or not (between-class). Since we expect that the visual features of the word stimulus play a crucial role, we also varied the degree of visual similarity of the reference stimulus to the target class. This was done by computing the cosine similarity between all word stimuli and selecting those that had the highest or lowest similarity respectively in the middle convolutional layer (excluding self-matches). See Figure 4 a) for a simplified visual representation of the procedure, and Fig-

ure 4 c) for example results for within and between-class ASM for each layer.

## Results

### Accuracy

After training the model on object and word stimuli only, the model was also tested on congruent and incongruent images. We expected a significant performance decrease for incongruent stimuli, because incongruent image and word features were expected to cause conflicting activations in the shared representations space (FC layers). Overall, the accuracy of the word recognition exceeded the accuracy of image recognition (see Figure 2), which was expected given previous literature (Rangari et al., 2023; Agrawal & Dehaene, 2024). Furthermore, the difference between congruent and incongruent stimuli points towards the predicted semantic interference effect. See Table 1 for exact accuracy values and Figure 2 for a visual representation of the results. In a simple feed-forward network, like the VGG 11, this difference should be explainable by the layer activations and hence we should observe a representational similarity (RS) between a) the object depicted on the incongruent image and the corresponding word-only image and b) the word depicted on the incongruent image and the corresponding object-only image, which interfere with each other.

Table 1: Testing accuracy across various conditions.

| Top | Img. | Wrd. | Con. | Inc. Img. | Inc. Wrd. |
|---|---|---|---|---|---|
| 1 | 75.0 | 94.7 | 71.7 | 53.8 | 4.1 |
| 5 | 90.2 | 96.0 | 87.8 | 74.9 | 9.2 |
| 5% | 95.2 | 96.8 | 94.5 | 87.7 | 21.3 |
| 10% | 96.3 | 97.2 | 95.9 | 91.3 | 30.9 |

An example was classified as correct if its output probability was among the top 1, 5, 5% or 10% highest probabilities after applying a softmax operation to the output layer.

### Representational Similarity Analysis

We computed the representational similarity (RS) between different conditions across multiple layers of the network model (see Figure 3). Each of those comparisons was baseline corrected, by computing the RS relative to the average of 1000 randomly selected stimuli (see Equation 2). Because object and word stimuli are visually very different, the absolute difference in RS is only little informative, for which reason we will mainly focus on the change profiles in RS across layers. This allows us to reveal the relative contribution of each layer to the overall formation of semantic representations throughout the network.

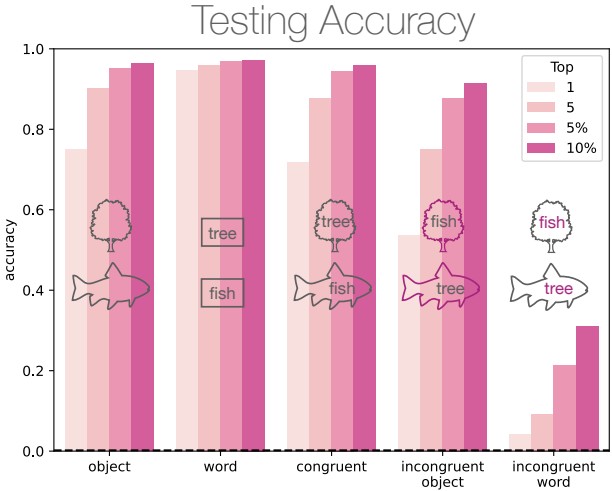

Figure 2: Testing accuracy across various conditions. *object*: ImageNet examples; *word*: word images; *congruent*: images overlaid with a word of the same class; *incongruent object*: images overlaid with a word from a different class, tested according to the object class; *incongruent word*: similar, but tested according to the word class. The dashed line represents the $\frac{1}{533}$ chance level. See Table 1 for an overview of accuracy values. An example was classified as correct if its output probability was among the top 1, 5, 5% or 10% highest probabilities after applying a softmax operation to the output layer.

**Objects and Words** Figure 3 a) compares the RS of object with word stimuli, whereas Figure 3 b) compares word to object stimuli (both relative to baseline). Those are expectedly almost identically and only differ in the baseline (due to the random selection), which acts as a sanity check. Objects and words of the same class do not expose a significantly higher RS when compared to any other class (baseline) in the middle and last convolutional layers. Within the FC layers, the same class object-to-word RS is elevated for the first and even more for the second FC layer. This suggests that common representations for objects and words emerge in deeper, fully connected layers of the network.

**Congruent overlays compared to Objects and Words** Figure 3 c) compares the RS of objects with congruent stimuli, whereas Figure 3 d) compares the RS of words with congruent stimuli (both relative to baseline). A slight increase in RS was observed in the last convolutional layer for the object-only to congruent stimulus comparison, with otherwise a somewhat flat profile. This indicates that the intra-class similarity of object features is lowest for the last convolutional layer (Yang et al., 2023). In turn, for the comparison between word-only and congruent stimuli, the RS increased steadily for the last convolutional and both FC layers. This can be explained by the fact that more visual features are shared between the congruent and object-only stimuli as compared

to the comparison between congruent and word-only stimuli in terms of number of pixels. Similar to the object-only to word-only comparison, the highest RS values have been found in the FC layers, which again indicates that object and word features are linked there. Since it is unclear, whether the object or word parts of congruent stimuli drive the effect, conclusive evidence that object and word features are indeed linked in the FC layers, can only be provided by the incongruent stimulus analysis.

**Incongruent overlays compared to Objects and Words** Figure 3 e) compares the RS of objects with incongruent stimuli, whereas Figure 3 f) compares the RS of words with incongruent stimuli (both relative to baseline). For the comparison with object-only stimuli (Figure 3 e), we observed a similar pattern as for the congruent condition, with an increased RS in the last convolutional layer and an otherwise flat profile. Again, this indicates elevated intra-class separability for visual object features (Yang et al., 2023). The comparison between the word-part of the incongruent stimuli with corresponding object-only images, in turn, revealed a similar pattern as for the object-to-word comparison (see Figure 3 a). This indicates that the word-part of the incongruent stimuli is processed independently of the object part at the feature level. Comparing incongruent to word-only stimuli (Figure 3 f) reveals a similar pattern as for the comparison between congruent stimuli and words (Figure 3 d). In turn, comparing the object part of incongruent stimuli to their corresponding words yields a similar activation profile as for the comparison between object and word-only stimuli (Figure 3 b). Especially the comparison between the word part of the incongruent stimuli to the corresponding object-only images (see Figure 3 e magenta line)and the comparison between the object part of the incongruent stimuli to the corresponding word-only stimuli (see Figure 3 f black line), point towards an independent object and word processing of both stimulus components at feature level. The highest RS was observed in the FC layers, which indicates that indeed the link between object and word representations are formed there. This potentially explains the semantic interference, which causes the decrease in accuracy for incongruent stimuli. To reveal how this link is formed, we conducted an analysis of activation similarity maximisation.

**Activation Similarity Maximisation**

To verify that the FC layers perform the translation of visual representations into joint semantic representations, we introduced activation similarity maximisation (ASM). This procedure is very similar to a regular activation maximisation (Erhan et al., 2009), which aims to generate a stimulus from random noise that maximises the activation of a target unit. Thereby, it is revealed what the network considers to be a prototypical example of a specific class. Here, we added a regularisation term, which aims to simultaneously maximise the RS in a given layer to a reference stimulus (a written word image). This means that the final generated stimulus is both

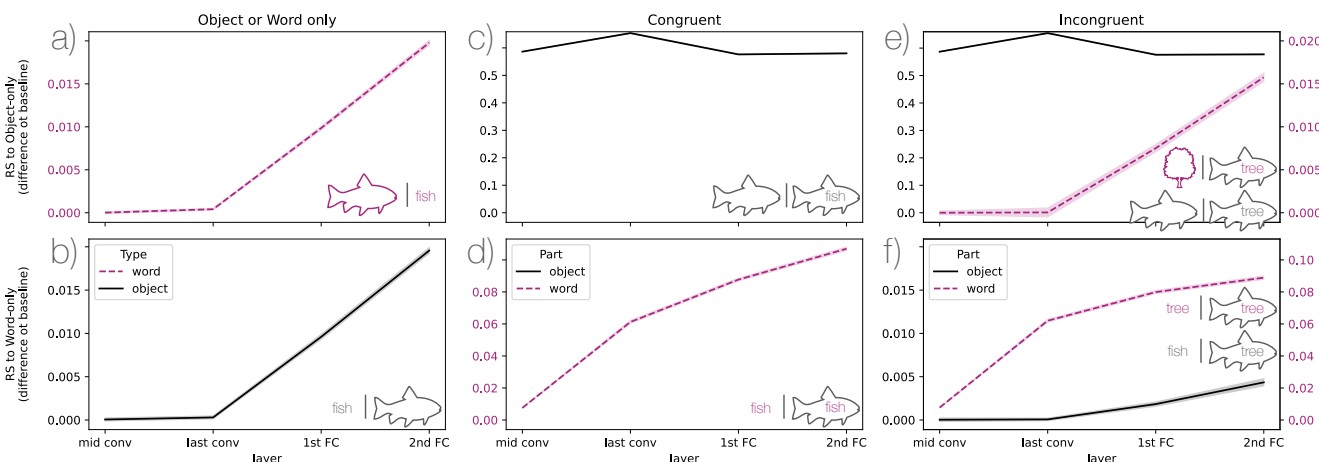

Figure 3: Representational Similarity (RS) relative to a baseline across different layers for various stimulus types: **(a)** objects and words; **(b)** words and objects; **(c)** objects and congruent stimuli; **(d)** words and congruent stimuli; **(e)** objects and incongruent stimuli; **(f)** words and incongruent stimuli. See also the illustrations. The x-axis denotes the network layers middle convolutional (mid conv.), last convolutional (last conv.) and fully connected layers ($1^{st}$ FC, $2^{nd}$ FC), while the y-axis indicates the RS values. Shaded areas represent the 95% confidence interval.

prototypical for the respective output class, but also activates a given layer similarly to a word-only stimulus (see Equation 3 and Figure 4 a). Then, the RS between the generated stimulus and the reference word stimulus were computed to reveal how close the word representation resembles the prototypical class representation in each layer. In addition, we varied the reference word stimulus in two stages. It could either be of the same class as the output unit (within-class) or a randomly selected different class (between-class). The between-class comparison was further varied such that the representation in the middle convolutional layer is either very similar to the representation of the word-only stimulus of the target class or very dissimilar (e.g. "goldfish" vs. "goldfinch" or "fly").

**Within and Between-Class ASM** We found that the RS in the *middle convolutional layer* between the generated and reference stimuli are very similar (see Figure 4 b left) for within and between-class comparisons. This indicates that no link between object and word features was established here. The examples depicted in Figure 4 c) show that word and object features are both visible simultaneously in the middle convolutional layers. In addition, the visual features are preserved such that they can be understood by humans. In other words, the features of the object and the visual word form co-exist in the middle convolutional areas (e.g. the target "goldfish" and the reference "fly"). The slight decrease in similarity in the middle convolutional layers for the five most dissimilar classes (see dark magenta line in Figure 4 b) left), most likely results from the fact that those examples were specifically selected to be most dissimilar in this layer.

Irrespective of the within or between-class comparison, we found an increase in RS between the generated and reference stimulus for the *last convolutional layer* relative to the middle convolutional layer (see Figure 4 b left). A similar increase in RS has been observed for the activations of word-to-word or object-to-object (see Figure 3), which most likely reflects the network's ability to differentiate between classes in general (Yang et al., 2023). However, since the increase in RS is different for the within and between-class comparison, we conclude that semantic information already interferes to some degree with the discriminative power of the network at this stage.

The RS between the generated stimuli and the reference words clearly separates the within and between-class comparisons in the *fully connected layers* (see Figure 4 b left). Remarkably, the final cosine similarity for the within-class comparison approaches 0.9 in the $2^{nd}$ FC layer, which indicates that the regularisation term (see Equation 3) only moderately impacted the final generated result. Because the activation evoked by the word-only stimulus in the $2^{nd}$ FC layer is very similar to the activation evoked by the generated stimulus, the word stimulus activation serves as a good approximation for the prototypical activation in that layer, which indicates that the word features themselves have resolved in general semantic class features. In turn, the RS for the between-class comparison is substantially lower within the FC layers. This indicates that the visual word form does not serve as a good approximation for the prototypical activation if the classes of the target unit and the reference word mismatch. Again, this points towards competing activations in the FC layers, indicating a semantic conflict.

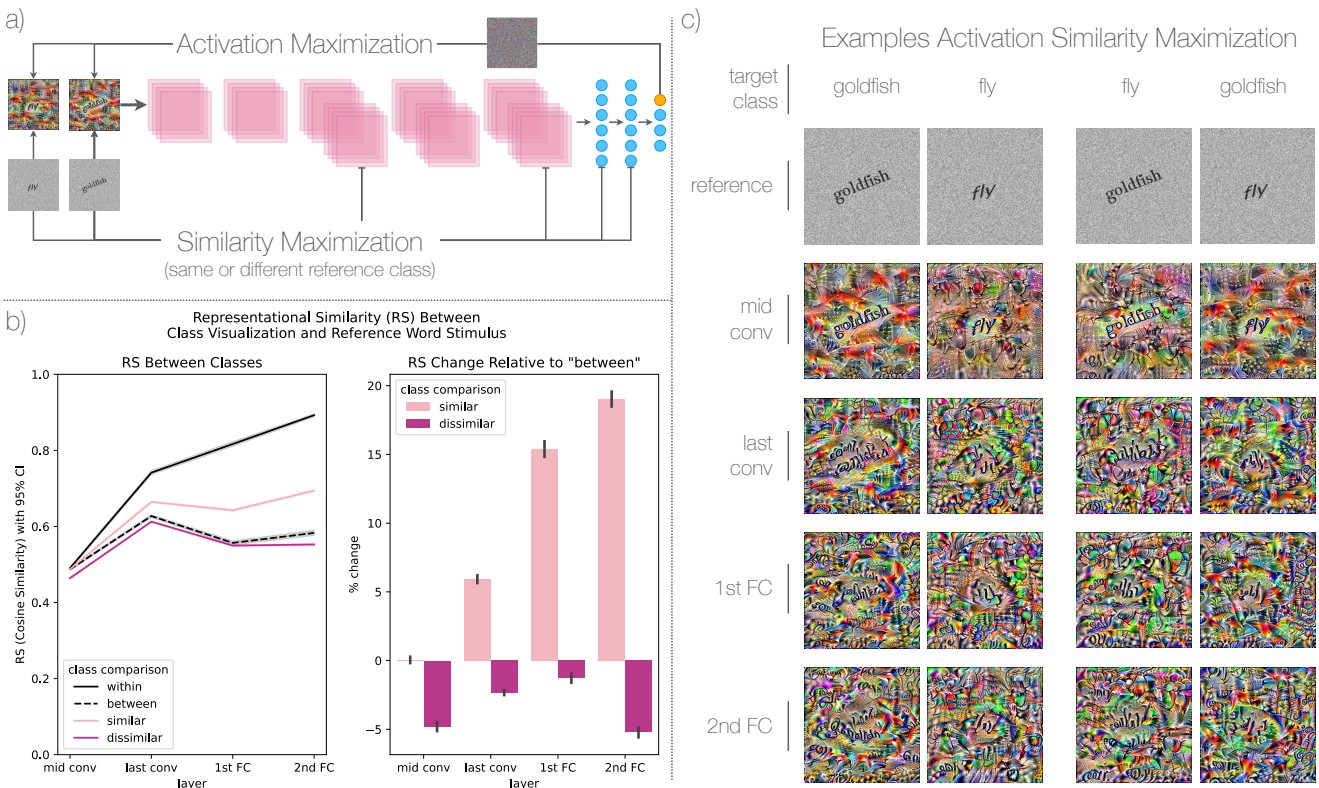

Figure 4: Activation Similarity Maximisation (ASM) framework and results. **(a)** Schematic of the ASM process: The model aims to generate a stimulus from white noise by optimising this stimulus such that the activation of a given output unit is maximised, and simultaneously the activation is as similar as possible to the activation of a reference word stimulus. This procedure was performed using a reference stimulus that is either the same or a different class as the maximised output unit. **(b)** Representational Similarity (RS) between the final result of the optimisation process and reference word stimulus: The left plot shows RS within and between classes across different layers. Thereby, the magenta lines differentiate between highly similar or dissimilar "between" classes. The right plot shows the percent change of RS relative to the "between" class comparison from the left panel for the top five visually similar or the bottom five visually dissimilar classes (percent change between dashed black line and magenta lines). **(c)** Examples of Activation Similarity Maximisation: Visualisations of the target class ("goldfish" and "fly") and reference stimuli across different layers (mid conv, last conv, $1^{st}$ FC, and $2^{nd}$ FC). The visualisations illustrate the result of the ASM procedure, depending on the layer at which the activation was compared for the similarity metric.

**Feature Similarity** Lastly, we investigated, whether the word features themselves contribute to the final RS in the FC layers, by varying the visual similarity of the between-class comparison, relative to the word representation corresponding to the target unit. Figure 4 b) right compares the RS for similar and dissimilar classes to the RS of all "between classes" (dashed black line in left panel) as the relative change in percent. For example, a class similar to the word "goldfish" would be "goldfinch", whereas a class dissimilar to the word "goldfish" would be "fly". Especially for the FC layers, an increase of RS of almost 20% could be observed for those reference words that are visually similar to the target class, while a significant decrease could be observed for reference words that are visually dissimilar to the target class. This indicates that the visual features of the reference word stimulus contribute to the final semantic representation in the FC layer. In other words, the semantic representations that are formed by the

network in the FC layers, are most likely projections of visual features into a high-dimensional representation space, as has been proposed previously (Haxby et al., 2020).

## Discussion

We demonstrated how semantic knowledge in a CNN emerges from the joint training of object and written word images. In a first step, we used a pre-trained VGG 11 CNN (Simonyan, 2014), which we then continued to train on ImageNet 1k visual objects (Deng et al., 2009) and written word stimuli created from the human-readable ImageNet labels. Pre-training thereby served as a proxy for a pre-literate infant that is already capable of naming different objects, and the joint training that followed thereafter, mimics the acquisition of reading. Using a picture-word interference task (Rosinski et al., 1975), we demonstrated that semantic interference caused a significant reduction of the network's accuracy, when

tested on object images, overlaid with incongruent words. A representational similarity analysis revealed that semantic interference occurs in the fully connected layers. Via activation similarity maximisation, we could further show that visual feature projections from the convolutional layers into the fully connected layers crucially impact the final semantic representations. We conclude that semantic representations are formed by projecting stimulus features into a shared space, similar to what has been proposed by theoretical frameworks (Haxby et al., 2020). In contrast to newer contrastive learning strategies such as SimCLR (Chen et al., 2020) or CLIP (Radford et al., 2021), traditional supervised learning methods might intuitively correspond better to human written language acquisition. Contrasting both (potentially against brain data) however, might reveal insight about the nature of representations that emerge given each strategy.

## Reduced Accuracy for Conflicting Information

Reduced behavioural accuracy in humans for conflicting information has been observed for the Stroop effect (Stroop, 1935) (see also: Laurienti et al. 2004), where colour words printed in incongruent font colours lead to increased error rates. Similar effects have been reported from the picture-word interference task (Rosinski et al., 1975), which has been proposed to be a variation of the Stroop effect sharing the same underlying computational principles (Starreveld & La Heij, 2017). Moreover, other multi-modal tasks involving high level conflicting stimuli, such as faces and names (Egner & Hirsch, 2005) or conflicting auditory and visual stimuli (Thomas et al., 2017), yield similar results. Since, our model was specifically chosen to only include feed-forward connections, this decrease in accuracy must stem from the visual features and their high level projections throughout the model. We hence assume that low level features had been projected into a shared representation space (Haxby et al., 2020), where conflicting concepts compete for the network's resources. Recent theoretical frameworks promote the idea of distributed semantic representations (Haxby et al., 2020), which is in line with our findings. However, due to methodological limitations, capturing these distributed patterns in neuroimaging data remains challenging (Frisby et al., 2023).

## Semantic Knowledge via Conjoint Feature Projections

We could demonstrate that semantic knowledge forms in the FC layers, which appear akin to the function of the left anterior temporal lobe (Mesulam et al., 2013; Li & Pylkkänen, 2021; Farahibozorg et al., 2022). Previous research showed that semantic interference is less pronounced if the interfering stimulus is similarly shaped as a target word (Rayner & Springer, 1986). This is in line with our finding that the semantic representation of a certain class is more similar to the representation of similar words. Agrawal & Dehaene (2024) demonstrated that CNNs trained on object and written word recognition develop activation patterns similar to the VWFA in the convolutional layers. This is most likely the result of the

network recognising reoccurring lexical patterns (e.g. "sh") which are similarly processed as reoccurring visual object features (e.g. eyes). The combination of visual word form features are then projected into the high-dimensional representation space. It has been shown that humans trained on new words that are similar to already existing representations (e.g. "BANARA" vs. "BANANA") suffer from decreased word identification performance (Bowers et al., 2005). Furthermore, it has been shown that word identification error rates increase when a centrally presented target word is flanked with visually similar distractor words (Vandendaele & Grainger, 2023). In line with our results, we interpret those findings such that the set of activated visual word form features, biases the final decision in the high-dimensional representation space.

## Limitations and Future Perspectives

The main limitation of our approach is the simplified architecture of the model. Residual and recurrent connections (He et al., 2016; Hochreiter, 1997) as well as overall more biologically plausible architectures such as CORnet-S (Kubilius, Schrimpf, Hong, et al., 2019) should be considered in the future. This requires analyses to be extended to the time domain as well. In addition, increasing structural plausibility can come at the cost of decreased performance for certain tasks (e.g. CORnet-Z Agrawal & Dehaene 2024), which needs to be taken into consideration and assessed for each model. Using a dataset like Ecoset (Mehrer et al., 2021), specifically designed to include the most common classes that humans encounter, might contribute to more realistic representations as well. In addition, it could be shown that introducing brain-like temporal dynamics can further improve the overall explanatory power of DNNs as a model for certain brain functions (Duecker et al., 2024). Future research should also incorporate the comparison to brain imaging data. A similar task as employed here could be used in an fMRI or MEG study. We predict that object and written word feature patterns can be distinguished using RSA in earlier visual cortex areas (e.g. V1 to V4), but not - or to a lesser extent - in more downstream regions like left anterior temporal lobe, where in turn the distinction between same and different class activation patterns become more prominent. Furthermore, lexical parafoveal processing in early visual cortex (Pan et al., 2021, 2024) as well as pre-saccadic feature processing in general (Fakche et al., 2024), might be the result of pre-activating visual word feature projections. Investigating the similarity between neural network activations of multi-modal models and human brain data could thus help to shed light on the overall computational principles of written language processing, which potentially extends to human multi-modal perception in general.

## Conclusion

The results presented here, indicate that semantic representations can be viewed as an emergent property of CNNs jointly trained on object and written word images. This can be observed on the behavioural level and the level of network representations. Since abstract semantic representations were

found to be limited to the FC layers, we conclude the acquisition of reading can be modelled as the process of projecting combinations of object and written word features into a shared high-dimensional space.

## Acknowledgments

This work was supported by a Wellcome Trust Discovery Award (grant number 227420) and the NIHR Oxford Health Biomedical Research Centre (NIHR203316) attributed to O.J.

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
