# OpenReview forum: "Decoding Semantics: A Multi-Modal CNN as a Model for Human Literacy Acquisition"
_ccneuro.org/CCN/2025/Proceedings — CCN 2025 Proceedings asProceedingsPoster_

### Official Review · Reviewer_geQK · 2025-03-31
**Excelent work**

**Soundness:** 3
**Clarity:** 3

**Comments:**

The paper address for how CNNs learn to encode semantic information from images (e.g. of a tree) and words (e.g. "tree"). Their findings shows that information from images and words are projected together and that incongruent information during training (e.g. showing an image of a fish next to the word "tree") can interfere with the learning process. Finally, they postulate that this methodology can help to better understand the human literacy acquisition.

Proposed improvements:

L205: Authors explain that they will only analyse the mid and last convolutional layers "Since early convolutional layers are expected to  only process relatively low level features" → It would be interesting to test this assumption

Table1 and Fig1: I was unable to find what 1, 5, 5%, and 10% means. Nor in the table/figure captions or the text.

**Expertise:**

2

**Interest:**

2

---

> ### Author Rebuttal · Authors · 2025-04-14
>
> We would like to express our sincere appreciation for the overwhelmingly positive feedback received from the reviewer. We have updated our manuscript according to the reviewers suggestions where necessary (blue text).
>
> The reviewer proposes testing whether early convolutional layers genuinely represent low-level features. We appreciate the reviewer’s insightful comment. In the current manuscript, our primary focus lies on the link between visual objects and visual word form representations, rather than delving into the intricacies of stimulus processing itself. As previously demonstrated (Zeiler & Fergus, 2014; Yosinski et al., 2015), early convolutional layers predominantly encode rudimentary patterns such as lines or corners. We did not presuppose any significant semantic link between images and words at this level, consequently, we opted to omit the rather computationally intensive analysis of early layers. Furthermore, we do not observe an interaction between image and word features for middle and high convolutional layers, leading us to assume that this phenomenon does not apply to low-level convolutional layers as well. Nevertheless, it would be intriguing to explore this link employing a neural network architecture that incorporates feedback connections. In such a scenario, top-down information could potentially exert a substantial influence on even low-level layers (Gilbert & Li, 2013). In order to make our reasoning more clear, we modified section “Representational Similarity analysis” within “Methods”, starting from line 197.
>
> The reviewer points out that the meaning of 1, 5, 5%, and 10% is unclear. We express our gratitude for bringing this to our attention and concur that an additional mention in the corresponding captions would be highly beneficial. Consequently, we have incorporated the information into Table 1 and Figure 1.

---

> > ### Comment · Reviewer_geQK · 2025-04-21
> >
> > I would like to thank the authors for the explanation about the exploration of early convolutional layers and the modifications to the manuscript.

---

### Official Review · Reviewer_xP3H · 2025-04-01
**Interesting results regarding semantic representations but some conceptual and methodological limitations to address**

**Soundness:** 1
**Clarity:** 2

**Comments:**

In this papers, the authors train a basic CNN (VGG11) to examine how humans might acquire semantic representations for reading and literacy, and learning to link images and visual forms for words. While I thought the paper was well-written and the analyses were well-conducted, there were a substantial number of conceptual and methodological weaknesses in this work that limit the impact of this submission. I hope the authors find my comments useful in addressing some of these concerns.

Weaknesses
- If I’m understanding correctly, the model is fine-tuned with either images from ImageNet, or Word-Images to predict 533 classes, with the same class prediction if either the image or the word-image is matching. However, isn’t this a very strong form of supervision to help link vision and language by linking them to the same class output unit? From a machine learning perspective, many of the findings from this work like the interference seem like they would ought to fall out naturally from this kind of direct supervision
- Furthermore, the authors claim that models like CLIP (Radford et al., 2021) or CVCL (Vong et al., 2024) are not good models for reading since the labels for reading are from the visual, not the language encoder. However, at least for CLIP, there is a lot of evidence that the model can read, such as examples that CLIP can suffer from “typographic attacks” when placing a written label of a different class on an object. The work from Goh et al. (2021) also showed evidence that CLIP has multimodal neurons in the vision encoder, that activate for both images and written text (in visual form) for a variety of concepts.
- In light of these two points, I’m not particularly sure that the findings reported here are particularly novel from what we already know from prior work around how semantic representations develop in neural networks. But in order to establish or falsify this, I would recommend the authors run their evaluations on a set of existing pre-trained vision models, including CLIP, and other standard supervised or self-supervised models. I would be much more convinced if only the kind of training done in this work (or with my additional suggestion below) achieves the pattern of results shown and not any existing vision models
- While the authors primarily investigate at which depth semantic representations emerge in a trained model, I think it would also be interesting to explore this phenomena over the course of training (either starting from an ImageNet pre-trained checkpoint, or from scratch)

Suggestions
- I’m surprised so many classes from ImageNet were filtered out. Is this due to the 10 character limit on classes, and how strong of a constraint is that? Sticking with the original 1000 classes might make it easier to evaluate standard pre-trained models (not fine-tuned on this dataset) more easily
- Another limitation is the use of single word labels. One suggestion (closer to CLIP) would be to take a image-caption dataset, convert the captions to visual forms like the approach in this paper, and use a contrastive learning approach like SimCLR (e.g. comparing the image vs. the visual caption), and seeing whether that learns like this model. I believe that might increase the external validity of this work, as such a model would need to learn to read in natural language, rather than labels

References
- Goh, et al., "Multimodal Neurons in Artificial Neural Networks", Distill, 2021.

**Expertise:**

2

**Interest:**

2

---

> ### Author Rebuttal · Authors · 2025-04-14
>
> We express our gratitude for the reviewer’s time and thoughtful comments on our manuscript. We have updated our manuscript accordingly (blue text).
>
> From a machine learning perspective, our findings appear as expected from direct supervision. While we hypothesised this outcome, we couldn't rule out the possibility of independent representations emerging. We could reveal that image and word representations remain distinct at the feature level, combining only after projection into fully connected layers. The interference stems from the link between visual form of objects and words. This approach offers a neuroscientific perspective on semantics through bottom-up features in a distributed, high-dimensional space (see e.g. Haxby et al., 2020).
>
> We acknowledge the apparent inconsistency between referencing CLIP while using supervised learning. Our primary goal was to model infant reading acquisition. Contrastive learning, which relies on positive/negative pairs, is less aligned with the natural learning process of reading. We clarified this in “Model Training” (line 117).
>
> Regarding our selection of only 533 ImageNet classes: we aimed to balance diversity of visual stimuli with standardisation. We introduced variation through different fonts, sizes, rotations, and shearing, but excluded words exceeding 10 characters to prevent cropping issues while maintaining font variability. We also excluded multi-word labels and those with special characters like dashes, again prioritising standardisation. Furthermore, we still believe that 533 classes do not trivialise the problem. While the performance of the network cannot directly be compared to a 1000 class network, the important factor here was to ensure "high enough" accuracy without overfitting. In retrospect, using Ecoset (Mehrer et al., 2021) might have reduced the number excluded labels. We modified “Methods” (line 142) and “Discussion" (line 575).
>
> The suggestion to use SimCLR, is highly appreciated, but doesn't align with our goal of emulating (supervised) infant reading acquisition. However, testing whether a CNN could link images and captions, and how word relatedness would be represented within the network's structure, remains very interesting. Since infants typically progress from single nouns to increasingly complex sentences, applying a similar training strategy to neural networks might yield interesting outcomes, though this extends beyond our current scope. We added this to the “Discussion” (line 504).

---

> > ### Comment · Reviewer_xP3H · 2025-04-20
> >
> > I appreciate the authors' attempts at clarifying some of the motivation and modeling choices in their revised paper, particularly around the choice of classes used, but similar to Reviewer YQq7, I believe these edits still do not address some of the conceptual and methodological concerns I have with this work, and still have some concerns with the work:
> >
> > First, the authors claim that their supervised approach is superior for emulating infant reading acquisition, as compared to the use of positive/negative pairs in other contrastive learning approaches. However, even with the updated "Model Training" section, it's still not exactly clear to me why this is a priori true, or what the evidence for using a set-up with such strong supervision comes from in a naturalistic setting. Where exactly are these supervised labels coming from in a naturalistic setting when a child is leading to read enabling them to link the visual vs. visual word-form of a category? (and maybe as a side question: how much of this is enabled or driven by the existing link between the visual and auditory forms of a word?) Perhaps one suggestion is to be more upfront about which level in Marr's hierarchy the work is aiming to capture? Is it more at the implementation-level (capturing the specific neural implementation for reading acquisition?), or the computational-level (capturing the macro-level patterns in infant reading acquisition?). The arguments presented against using a contrastive approach suggests the former, but similar to the concerns from Review YQq7 the authors seem hesitant to explore models like CORnet which are more biologically plausible.
> >
> > Second, I would still like to see more of a discussion on the specific developmental trajectory of infant reading acquisition. What are the major milestones and specific phenomena that take place as infants go from knowing how to see, to knowing how to read? More exposition here would help explain why interference effects are the main measure, but it would also be interesting to have a better sense at what age they begin to show these effects.
> >
> > Third, my more general point surrounds the logical argument in this work. While the approach used in this work: jointly training visual and word forms via supervised learning, demonstrates the kinds of interference patterns that learning to read would produce, is this the only training set-up that reliably does so? In essence, the paper provides confirmation for this effect for one type of model and training regime, but it would be a much stronger paper if it were able to rule out or falsify that other models (e.g. CORnet) or training approaches (e.g. contrastive learning) do not reliably lead to the same patterns of behavior, e.g. if they displayed independent representations emerging. My comments around CLIP were less about the specific contrastive training method, but that its representations and behavior (e.g. from typographic attacks) really does seem to show the same kinds of reading interference effects present in the models from this work, which intuitively suggests to me that there is a much larger class of models where visual and visual word-forms can be semantically linked rather than the specific approach described in this work. I think that a much better version of this work would characterize the kinds of pre-training, data mixes, architectures, and training supervision produce these, rather than opting for a single model choice and arguing that this is the way it ought to be without these comparisons.

---

> > > ### Author Response · Authors · 2025-04-20
> > >
> > > We thank the reviewer for engaging in the discussion. For many parts, we would like to refer to the "Reply Rebuttal Comment" originally addressed to reviewer YQq7.
> > >
> > > We agree that classic supervised learning (SL) does not provide a one-to-one mapping to the process of infant reading acquisition. However, contrastive learning (CL) in our opinion maps conceptually worse than supervised learning. As the reviewer pointed out correctly, human reading acquisition in most cases is accompanied by auditory forms as well. While this could already be interpreted as some form of "labelling", we derived our strategy mostly from the process of how reading is often taught to infants: Children do receive consistent "labelling", e.g. by showing visual representations of words and corresponding objects in schooling material (e.g. a picture of a dog next to the word "dog"). While this does not exactly match SL in a deep learning context, in our opinion SL resembles this learning process better than CL (see also response to reviewer YQq7).
> > >
> > > Regarding the choice of architecture, please also see our response to reviewer YQq7. It could be demonstrated that VGG-style models produce representation patterns equally or even more similar to human brain response patterns as compared to the CorNet family (Nonaka et al., 2021) and thus show higher biological plausibility in some aspects. In addition, CorNet-style networks lack "behavioral plausibility" in that classification accuracy for images is significantly reduced as compared to other CNN architectures. Since our task (interfering multi-modal stimuli) requires high initial classification accuracy, which furthermore translates better to human level performance, we opted for a model (VGG11) which produces equal or more biologically plausible representations, while also producing more plausible behavioral responses (more human-like accuracy), even if less anatomically motivated. We agree with the reviewer that comparing multiple CNN architectures (not limited to VGG or CorNet) and training strategies on a similar task might reveal additional insights, especially if compared against actual brain data. This however, would go far beyond the current scope of this manuscript. Here, we propose a model that views the link between visual word and object representations as the consequence of feature projections into a high dimensional representation space by utilizing a single encoder. We do not argue that VGG11 is generally better suited for model-to-brain comparisons, nor do we wish to claim that SL produces more brain-like representations than CL. Instead, our main goal was to utilize a model that produces somewhat comparable brain-like representations, while at the same time showing relatively high behavioral plausibility. VGG11 (and SL) covers both aspects and also allows for a relatively straight forward investigation of layer representations, which was the main goal of our study to demonstrate how a joint encoder can lead to joint representations.
> > >
> > > Reproducing every stage of infant learning (e.g. from vision, via auditory training to single word and complex sentence reading) extends beyond the scope of this work. We do however, acknowledge that this would potentially provide much more insight into infant reading acquisition than our current manuscript. For this a collaboration with an infant laboratory would certainly be very fruitful. Our manuscript could provide the basis for the hypothesis that semantic interference between objects and written words occurs at early school age, which should be accompanied by the respective changes in brain response patterns.
> > >
> > > We also agree with the reviewer that typographic attacks can be produced using different training strategies (and potentially model architectures). While CLIP can produce similar behavioral patterns, we complement this kind of research by the finding that CL is not a requirement to produce interfering concepts or to link multiple modalities. We fully agree that contrastive learning (CL) is a powerful framework for learning shared representations, but the fact that the absence of contrastive loss also produces meaningful multi-modal representations seems striking to us. More important however, our work does not focus solely in the interference of concepts, but also demonstrates that the interference in our model most likely stems from visual features that project into a shared space. This support relatively recent theories and empirical findings that point towards a distributed "vectorized" organization of knowledge in the brain (Haxby et al., 2020; Vidal et al., 2021; Contier et al., 2024; Piantadosi et al., 2024). Those theories would predict that knowledge can be seen as a "vector" of features that conjointly form a concept. Our finding that very distinct sets of activated features (from objects and words) can lead to an activation of the shared representation space that leads to conceptual interference, supports this concept.

---

### Official Review · Reviewer_YQq7 · 2025-04-01
**An ambitious paper on semantic representations that suffers from exaggerated claims**

**Soundness:** 1
**Clarity:** 1

**Comments:**

The paper compares the representation of images and words related to a given concept, obtained from a convolutional network. It uses RSA as a tool to quantify pattern similarity.
As expected, the network representations seem to be driven by semantics in the last, fully connected layers.

The story is then embedded in a rather unconvincing conceptual framework, where the CNNs are interpreted as a brain model, and the authors of the paper seem to believe that they are allowed to comment on the relevance of conflicting cognitive theories on the basis of this experiment. I fail to see how they can do this.

* First, as the authors note, the natural tool for this kind of analysis would be contrastive learning. Why they have not done so remains a mystery. What is the reason ? Using CNNs instead is not a good idea. If the authors wanted a more biologically plausible model, CorNET would have been a better choice.
 Why would "words themselves [be] linked to the visual representation of the named objects" in a CNN?

* The paper makes heavy use of vague statements to justify methodological choices, e.g. "a substantial performance penalty", "somewhat comparable".

* Noise addition on the word images seems awkward * Inconsistency in eq. (1)-(2): (A,B) RS_l^{BL(A,B)} depends only on A, not on B !

* Fig. 3 is hard to parse and gives ambiguous information because the y-scale is jittered from one plot to the next.

* I did not get the message from the "optimal stimulus" reconstructions in Fig. 4. The fact that goldfish and fly give the same output in the FC layers means that the model did not capture anything (?)

* What do the authors mean by "semantic representations in a CNN emerge **naturally** by a joint training of object and written word images"?

* "we conclude the acquisition of reading can be modelled as the process of projecting combinations of object and written word features into a shared high dimensional space." I think that this conclusion is vague enough to be acceptable, yet not well supported by the presented work.

**Expertise:**

3

**Interest:**

2

---

> ### Author Rebuttal · Authors · 2025-04-14
>
> We appreciate the reviewer’s investment of time in providing feedback on our manuscript. We have updated our manuscript accordingly (blue text).
>
> The reviewer suggests that contrastive learning may be preferable to CNNs. However, using a CNN architecture is not mutually exclusive with contrastive learning. Our approach models literacy acquisition in infants. Contrastive learning, which relies on positive/negative pairs, is less aligned with the natural learning process of reading. We clarified this in “Model Training” (line 117).
>
> The reviewer mentions CorNET as a more biologically plausible alternative. We had referenced CorNET-Z, a CNN variant, and noted its low accuracy (36.8% on ImageNet; Agrawal & Dehaene, 2024). Given the importance of high image classification accuracy in our study, we selected VGG11. Additionally, VGG models perform well in predicting brain data (Nonaka et al., 2021), with VGG variants outperforming CorNET models on brain similarity scores. We clarified this in “Network Architecture” section (line 97) and expand in the “Discussion” (line 568).
>
> The reviewer questioned the added noise to word images. This was based on previous work (Van Vliet et al., 2022; corrected citation) and serves to simulate environmental variation and better match test conditions. We clarified this in the “Model Training” section (line 132).
>
> Concerning equations (1) and (2), the reviewer notes the baseline relies only on A. We clarified this by specifying that baseline similarity (for A vs. B) is calculated using randomly sampled class examples C (line 212) to relate with A, and RS is computed as the difference from this baseline (line 217).
>
> We addressed ambiguity in Figure 3 by re-scaling the y-axes.
>
> The reviewer inquires about the meaning of “semantic representations in a CNN [that] emerge naturally by a joint training of object and written word images.” We define it as a shared representation between visual objects and word forms. Through joint training using a single output layer, the model forms such representations without enforced modality separation, unlike prior work (Agrawal & Dehaene, 2024). We reworded the “Conclusion” (line 599).
>
> Lastly, the reviewer suggests we overstate our theoretical implications. We would like to clarify that our findings support the framework of distributed semantic representations (Haxby et al., 2020) without commenting on other cognitive theories. We revised the relevant “Discussion” section (line 526) accordingly.

---

> > ### Comment · Reviewer_YQq7 · 2025-04-19
> >
> > Thanks for opening the discussion. I note several clarifications by the authors, which certainly improve the paper.
> > However, I am afraid that the core issues are left as they are. The authors mostly restate (somewhat more clearly) the arguments made in the original submission, and the changes are actually very minor.
> >
> > * I don't understand why the authors reject contrastive learning. AFAIK cognitive neuroscience is not strong enough for such positions. To me this is a limitation of this work that needs to be acknowledged as such.
> > * I don't understand why the authors take the lower performance of CORNet as a motivation not to include it. If their interest is in finding a model of human reading, I don't think performance is a major goal.
> > * Justifying noise addition by the fact that it has been done in previous work also falls short as a justification.
> >
> > To summarize, while I agree that the paper is better than the first draft, I still think that there is too much of a gap between i) the current state of knowledge of the domain and the hypotheses used by the authors ii) the experimental results and the conclusions, and I am not compelled.

---

> > > ### Author Response · Authors · 2025-04-20
> > >
> > > We thank the reviewer for engaging in the discussion and look forward to potential discussions during the conference.
> > >
> > > We agree that contrastive learning (CL) is a powerful framework for learning shared representations and that it has been successfully utilized in cognitive neuroscience research. Our decision against CL was not motivated by a theoretical consideration that assigns more plausibility to classic supervised learning as opposed to CL per se. Rather, our decision was motivated by the methodological consideration of utilizing a shared architecture that does not force a shared representation space and investigate whether such shared representations would emerge regardless. In fact, the absence of contrastive loss makes it all the more striking that an alignment of visual word and object features emerges at all. This alignment even causes interference with the model's decision process and seems to be the result of visual word and object features projecting into the same high level representations. While CL is often used as a tool to force those representations, we could demonstrate that classical supervised learning can lead to similar results. However, we agree with the reviewer that directly comparing learned representations from supervised learning and CL - potentially against brain data - would provide the basis for an interesting follow-up experiment. Lastly, utilizing classic supervised learning is not uncommon for related work (See e.g. Hannagan et al., 2021; Agrawal & Dehaene, 2024; Agrawal & Dehaene, 2025; Van Vliet et al., 2025).
> > >
> > > We fully agree with the reviewer that not raw model performance, but rather brain similarity, should be the criterion for model selection when the goal is to approximate biological mechanisms. Our choice of a VGG-style model was motivated by its empirically observed higher correlation with human ventral stream responses (Nonaka et al., 2021). We acknowledge that CorNet models have been designed to be biologically grounded in architecture, but the variants that we considered show lower neural predictivity. In addition, our stimuli and task (interference from incongruently combined stimuli), requires a relatively high "raw" classification performance for both words and objects. Furthermore, the accuracy produced by VGG models is much more similar to what we expect from humans behavioral performance. Investigating the relationship between object and word representations, in our opinion, requires an alignment of behavioral performance as well to improve plausibility of the model choice. CorNet lacks this "behavioral plausibility". In sum, we chose to opt for a model that was shown to produce equal or even better representations as CorNet (when compared to human brain data) and provides more plausible behavioral results, even if less anatomically motivated.
> > >
> > > We added noise to our word stimuli to simulate the variability encountered in real world reading (different lighting conditions, backgrounds, poor print quality, etc.). Our justification therefore does not rely solely on referencing previous work.
> > >
> > > In summary, we acknowledge that we approached the presented topic from an angle that the reviewer disagrees on. However, we would like to emphasize that our work and interpretations are grounded in previous theoretical and empirical work (Haxby et al., 2020; Vidal et al., 2021; Contier et al., 2024; Piantadosi et al., 2024).
> > >
> > > References:
> > > Agrawal, A., & Dehaene, S. (2024). Cracking the neural code for word recognition in convolutional neural networks. PLOS Computational Biology, 20(9), e1012430.
> > >
> > > Agrawal, A., & Dehaene, S. (2025). Dissecting the cortical stages of invariant word recognition. bioRxiv, 2025-03.
> > >
> > > Contier, O., Baker, C. I., & Hebart, M. N. (2024). Distributed representations of behaviour-derived object dimensions in the human visual system. Nature Human Behaviour, 8(11), 2179-2193.
> > >
> > > Hannagan, T., Agrawal, A., Cohen, L., & Dehaene, A. S. (2021). Emergence of a compositional neural code for written words: Recycling of a convolutional neural network for reading. Proceedings of the National Academy of Sciences, 118(46), e2104779118.
> > >
> > > Haxby, J. V., Guntupalli, J. S., Nastase, S. A., & Feilong, M. (2020). Hyperalignment: Modeling shared information encoded in idiosyncratic cortical topographies. elife, 9, e56601.
> > >
> > > Piantadosi, S. T., Muller, D. C., Rule, J. S., Kaushik, K., Gorenstein, M., Leib, E. R., & Sanford, E. (2024). Why concepts are (probably) vectors. Trends in Cognitive Sciences, 28(9), 844-856.
> > >
> > > van Vliet, Marijn, Oona Rinkinen, Takao Shimizu, Anni-Mari Niskanen, Barry Devereux, and Riitta Salmelin. "Convolutional networks can model the functional modulation of the MEG responses associated with feed-forward processes during visual word recognition." (2025).
> > >
> > > Vidal, Y., Viviani, E., Zoccolan, D., & Crepaldi, D. (2021). A general-purpose mechanism of visual feature association in visual word identification and beyond. Current Biology, 31(6), 1261-1267.

---

### Meta-Review · Area_Chair_S37F · 2025-05-08

**Ccn Recommendation:** Accept as Proceedings

**Metareview:**

The AC agrees with the reviewers that the topic of this work would be of interest to the CCN community and is also impressed by the authors responsive rebuttal. While reviewer xP3H in particular has raised valid concerns, the authors have addressed most of them and the AC is inclined to agree that some of the important extensions suggested by the reviewer are beyond the scope of the present paper.

One qualm, however, that the AC would like to note, beyond what was already discussed in the reviews, is that the paper only refers to CORNet-Z, overlooking later extensions such as CORNet-S, that do perform well on both brain and classification benchmarks (see reference below).

While the reviews are mixed, the AC is inclined to think that this paper merits publication as a full CNN proceeding paper because it offers an interesting new approach to an important question (reading acquisition), and the methodology and results appear to be technically sound (especially after the authors' responses and revisions). The AC hopes, however, that the authors will correct they way CORNet is addressed in the paper.

Reference:
Kubilius, Schrimpf, et al. (2019). Brain-like object recognition with high-performing shallow recurrent ANNs. Advances in neural information processing systems, 32.

**Summary:**

The reviewers agree that the paper would be of interest to the CCN community, and have mixed views on the strengths of the results. Reviewer geQK expressed strong support for this work, while reviewers YQq7 and xP3H are more skeptical and raised several concerns about the choices of models, the comparisons between SL and CL and the validity of the training setup. Reviewer xP3H in particular mentioned a few important constructive suggestions such as looking at developmental trajectories of infant reading acquisition and exploring more systemically a broader range of models and training settings. The authors, in return, acknowledged the importance of these extensions of the work but argue that they are beyond the scope of the present paper.

**Expertise:**

2